# A Look Behind the Scenes at COVID-19: National Strategies of Infection Control and Their Impact on Mortality

**DOI:** 10.3390/ijerph17155616

**Published:** 2020-08-04

**Authors:** Samir Haj Bloukh, Zehra Edis, Annis A. Shaikh, Habib M. Pathan

**Affiliations:** 1College of Pharmacy and Health Science, Department of Clinical Sciences, Ajman University, PO Box 346 Ajman, UAE; s.bloukh@ajman.ac.ae; 2College of Pharmacy and Health Science, Department of Pharmaceutical Sciences, Ajman University, PO Box 346 Ajman, UAE; 3Advanced Physics Laboratory, Department of Physics, Savitribai Phule Pune University, Pune 411007, India; annisshaikh786@gmail.com (A.A.S.); pathan@physics.unipune.ac.in (H.M.P.)

**Keywords:** COVID-19, SARS-CoV-2, human coronavirus, pre-existing conditions, public health strategy, prevention, containment, control measures, vitamin D, droplets, indoor and outdoor climate, post-lockdown period, recommendations, UAE, India

## Abstract

(1) Background: The severe acute respiratory syndrome coronavirus 2 (SARS-CoV-2) began spreading across the globe in December and, as of 9 July 2020, had inflicted more than 550,000 deaths. Public health measures implemented to control the outbreak caused socio-economic havoc in many countries. The pandemic highlighted the quality of health care systems, responses of policymakers in harmony with the population, and socio-economic resilience factors. We suggest that different national strategies had an impact on mortality and case count. (2) Methods: We collected fatality data for 17 countries until 2 June 2020 from public data and associated these with implemented containment measures. (3) Results: The outcomes present the effectiveness of control mechanisms in mitigating the virus for selected countries and the UAE as a special case. Pre-existing conditions defined the needed public health strategies and fatality numbers. Other pre-existing conditions, such as temperature, humidity, median age, and low serum 25-hydroxyvitamin D (25(OH)D) concentrations played minor roles and may have had no direct impact on fatality rates. (4) Conclusions: Prevention, fast containment, adequate public health strategies, and importance of indoor environments were determining factors in mitigating the pandemic. Development of public health strategies adapted to pre-existing conditions for each country and community compliance with implemented policies ensure the successful control of pandemics.

## 1. Introduction

A novel coronavirus (CoV) was identified in December 2019 in Wuhan, China. This highly contagious and previously unknown pathogen caused complications in the respiratory system and was named severe acute respiratory syndrome coronavirus 2 (SARS-CoV-2). WHO first declared the outbreak of coronavirus disease 2019 (COVID-19) a six public-health emergency of international concern, before subsequently declaring it a pandemic [1]. Human coronaviruses (HCoVs) are enveloped, non-segmented, positive-sense, single-strand ribonucleic acid (RNA) viruses [2]. HCoVs can infect animals and humans, causing respiratory, hepatic, and neurologic diseases [3]. CoVs are divided into four genera: *alpha-coronavirus* (α), *beta-coronavirus* (β), *gamma-coronavirus* (γ), and *delta-coronavirus* (δ) [2,3,4,5]. Human coronaviruses were identified in late 1960 and are known to infect humans, other mammals, and birds. To date, six HCoVs have been categorized. In 2003, a virus was identified in Guangdong province in China causing severe acute respiratory syndrome (SARS). Later, the virus was confirmed as a member of the beta-coronavirus family and was named SARS-CoV [6]. A decade later in 2012, two Saudi Arabian nationals were identified to be infected with another coronavirus. This virus belongs to the *beta-coronavirus (β)* family and was termed the Middle East respiratory syndrome coronavirus (MERS-CoV). In 2019, a novel CoV caused the outbreak of a SARS-like illness in the animal market of the Chinese city of Wuhan and was termed severe acute respiratory syndrome coronavirus-2 (SARS-CoV-2) [7].

The source of origination and transmission are relevant factors for the development of preventive strategies and treatment protocols. SARS coronaviruses may have circulated in humans before causing the outbreak in 2003 [8]. *Rhinolophus* bats were identified as having anti-SARS-CoV antibodies, thus suggesting the bats were a source of the virus [9]. The Middle East respiratory syndrome (MERS) coronavirus (2012) in Saudi Arabia is a beta-coronavirus originating from camels as the primary host [10]. In the case of SARS-CoV-2, only bats were identified as key reservoirs [11,12]. COVID-19 originated in Wuhan and was carried through migration by the population to different cities and other provinces around Hubei [12]. Infection rates increased rapidly and were controlled by a rapid and unprecedented lockdown of cities in the Hubei province followed by further control mechanisms for the whole country [12,13]. This unparalleled approach effectively prevented an exponential increase of infection on a national basis in China [12,13]. Similar measurements were implemented by many other countries once cases started appearing within their borders. Enforcement methods and periods of implementation varied from country to country and delivered mixed results. Factors influencing transmission, disease progression, susceptibility, and virulence are surfacing slowly [2]. Park et al. studied clinical and demographic characteristics of 44,672 COVID-19 cases in China and noted that everyone without age limitation is susceptible [14]. The fatality rate is highest in elderly, immunocompromised individuals with comorbidities [14]. The mean or median incubation time of COVID-19 is less than 13 days and should perhaps be considered to be lower than 4–6 days, similar to SARS-CoV and MERS-CoV median incubation times of 4.4 and 5.5 days, respectively [14]. SARS-CoV-2 is highly contagious and infection may occur by respiratory droplets carrying the virus or through contact with infected surfaces (formites) and subsequently touching mouth, nose, and eyes [15]. Liao et al. reported that particles smaller than 1 μm remain in the air for an unknown time and particles larger than 5 μm settle within one hour [16]. Particles with sizes larger than 10 μm, also called droplets, settle much more quickly and cause infections through fomites [16]. Thus, airborne transmission happens mainly via smaller particles. Fine particles smaller than 5 μm have the highest virulence factor because they remain in the air longer and can easily reach the lower respiratory tract [16]. Particles with a size larger than 5 μm can reach the upper respiratory tract [16]. Hou et al. showed that the nasal cavity is the first entry and proliferation point of airborne particles [17]. Accordingly, they argue, preventive measures such as wearing masks and using complementary therapeutic strategies on the nasal site may help control the spread of SARS-CoV-2 [17].

In this work, we analyzed the effectiveness and impact of national strategies and their impact on fatality rates by evaluating data of 17 countries until 2 June. We also investigated the public health strategies in the UAE, which has not been previously represented in the literature. We examined the relationship between pre-existing conditions, climate, population size, mean age, and level of SARS-CoV-2 control by implemented measures.

## 2. Materials and Methods

In this study, we analyze confirmed COVID-19 cases and fatalities in selected countries using the data on the worldwide geographical distribution of SARS-CoV-2 as of 2 June 2020 from the European Centers for Disease Control and Prevention (ECDC) website on 3 June 2020 [18]. For the first time, we highlight the situation in the United Arab Emirates (UAE) by discussing its national strategies and their impact on case count and mortality. Furthermore, we report the experiences of USA, Italy, and India as special cases. We compare several countries with similar population size in terms of COVID-19 case count and mortality. Sweden–Switzerland–Portugal, Germany–Turkey, and France–Spain are compared as examples of European countries. Latin America is represented by analysis of data from Brazil, Peru, Ecuador, and Mexico. South Korea and Japan were chosen as examples of Far Eastern countries with previous experience of SARS and efficient national strategies to combat COVID-19. We depict the results of the controversially discussed factors of average serum (25(OH)D) concentrations and median age, in addition to geographical factors of temperature and humidity. The outcomes of the analysis underline the importance of pre-existing conditions (health care system preparedness, median age, population size, population density, geographical conditions) and adequate public health strategies within a suitable time-frame. We comment on the effectiveness and problems of the implemented control mechanisms, including the general lockdown- and post-lockdown implications. Finally, we conclude with recommendations for the COVID-19 outbreak and future pandemics.

## 3. Results and Discussion

Progression of CoV to severe pneumonia is directly related to advanced age, history of metabolic syndrome, smoking, and other chronic conditions such as cardiovascular disease [19]. Countries with a high percentage of advanced age and high population density in urban areas need to impose swiftly efficient measurements to reduce fatalities [19]. Primc et al. analyzed the public health measures of European countries during the pandemic [19]. They concluded that qualitative development of the health care system and improving protective health conditions such as serum (25(OH)D) concentrations may be important for the control of COVID-19 [19]. To investigate the importance of serum vitamin D levels, median age, temperature, and humidity we compare infection control measures and their impact on COVID-19-related fatalities in Portugal, Sweden, and Switzerland (Figure 1).

Figure 1 shows the highest fatality numbers in Sweden, followed by Switzerland and then Portugal. Sweden is one of the few European countries refraining from nationwide closures and lockdown [19]. This, in combination with a lack of clear recommendations for the public in how to prevent viral spread, especially in indoor environments, may have aggravated SARS-CoV-2 spread in Sweden.

Primc et al. reported a possible relationship between low 25-hydroxyvitamin D (25(OH)D) concentrations and susceptibility to COVID-19 [19]. Vitamin D deficiency is prevalent in the northern hemisphere due to reduced solar irradiation especially in the winter months [20,21,22,23]. The infection rates of COVID-19, which started in the month of December 2019, are now showing exponential growth in Northern Hemisphere countries, [19,24,25]. According to a recent review by Grant et al., 25-hydroxyvitamin D (25(OH)D) concentrations may play a relevant role in the progression of COVID-19 [20]. Sufficient plasma (25(OH)D) concentrations around 75 nmol/L ensure cellular immunity, decrease the risk of infections caused by microorganisms, and protect against metabolic syndrome and other chronic illnesses [20]. Severe vitamin D deficiency is defined as serum 25(OH) D levels lower than 30 nmol/L [21]. Factors such as screen-based entertainment, sedentary lifestyle, and less outdoor physical activity are directly related to poor endogenous vitamin D synthesis [21,22,23]. Higher susceptibility of advanced age individuals to COVID-19 may confirm this hypothesis. Commonly, serum 25-hydroxyvitamin D (25(OH)D) concentrations are less than 30 nmol/L in elderly individuals and geriatric patients due to indoor dwelling, physical inactivity, and increased pharmaceutical drugs intake, especially in the winter [20,21,26]. Sufficient serum vitamin D levels above 70 nmol/L may have a positive impact on recovery rate and survival of patients during illness [20,21,22,23]. Ilie et al. associated mean serum vitamin D levels in elderly populations of European countries with COVID-19 infection and mortality [27]. They compared the mean serum vitamin D levels for the populations in Portugal (39 nmol/L), Spain (42.5 nmol/L), Switzerland (46 nmol/L), UK (47.4 nmol/L), Italy (50 nmol/L), Germany (50.1 nmol/L), Turkey (51.8 nmol/L), France (60 nmol/L), Sweden (73.5 nmol/L), and other countries [27]. Portugal has the smallest, Switzerland represents the average, and Sweden the highest mean serum 25-hydroxyvitamin D (25(OH)D) levels and similar population sizes of 10,281,762, 8,516,543, and 10,183,175, respectively (Figure 1) [18,27]. The highest number of fatalities is in Sweden, followed by Switzerland, and then Portugal, although the opposite trend was expected due to mean serum vitamin D levels and population number. We can conclude that mean plasma 25-hydroxyvitamin D (25(OH)D) levels have no direct impact on fatality rates and remain controversial in their relation to COVID-19. The results of Figure 1 show that the highest COVID-19-related mortality was in Sweden, followed by Switzerland and Portugal with median ages of 41.1, 42.7, and 44.6, respectively [24]. Therefore, we also can conclude from our analysis that median age is not a sufficient variable to predict fatality.

According to Scafetta, temperature and relative humidity have an impact on viral transmission [24]. Temperatures from 4 to 12 °C and relative humidity of 60–80% with gentle wind speeds increase the susceptibility to respiratory tract infections and the spread of SARS-CoV-2 [24]. Our analysis reveals that temperature and relative humidity are not the only factors with direct impact on fatality rates. The first fatal cases were announced in each of the three countries in March, while the first confirmed COVID-19 case was reported in Portugal, Switzerland, and Sweden on 3 March, 26 February, and 1 February 2020, respectively [18,24]. The average temperature in March 2020 was the highest in Portugal at 12.4 °C, followed by Switzerland with 2.2 °C, and the lowest was Sweden with −3.3 °C [24]. The lowest relative humidity was recorded in Portugal with 70%, Switzerland’s was 73%, and the highest was in Sweden with 88% [24]. According to these numbers, the spread of COVID-19 should be higher in Portugal, but again it was Sweden. An explanation for the higher fatality in Sweden is the cold weather itself. The virus cannot survive for long outside in the cold, but spreads easily because people remain indoors [24]. Indoor air is more suitable for the spread of SARS-CoV-2, especially due to close contact of people in unventilated, closed rooms without UV irradiation [24].

A coordinated effort to educate the population about the means of preventing transmission and infection in indoor and outdoor environments during a pandemic is crucial to avoid uncontrolled spread of the virus. Indoor and outdoor climate are related to each other and aggravate viral transmission under given conditions. Indoor environments can proliferate SARS-CoV-2 infection by raised viral loads through droplets and direct contact on infected surfaces more than an outdoor climate. This also confirms the spread of COVID-19 in household contacts and the need for indoor social distancing. According to Luo et al., pre-symptomatic cases within the household may transmit the virus rapidly [28]. Especially vulnerable and high-risk individuals are at risk of being infected. Luo et al. concluded in their prospective cohort study that the risk of transmitting COVID-19 is highest in household contacts [28]. Public health strategies need to include habitancy and dwelling patterns in relation to climate in measures to control outbreaks. Including dwelling modes in virus containment and control is essential, especially if nationwide closure and lockdowns are not imposed. In addition, a mask wearing strategy needs to be enforced in indoor and outdoor settings, especially in countries with a high population density. Wearing masks can reduce the airborne transmission by droplets and smaller particles, especially in indoor environments [16,17]. A study compared community-wide mask compliance in relation to the number of confirmed SARS-CoV-2 cases/fatalities in Hong Kong, Singapore, and other countries [29]. Singapore has the second highest population density in the world, directly followed by Hong Kong [29]. Due to their previous experience with the SARS virus in 2003, 97% of the studied population in Hong Kong began using masks when the first COVID-19 confirmed case emerged [29]. This mask-wearing strategy combined with social distancing, personal hygiene, cancellation of social gatherings, use of the home office, and school closures resulted in the effective control of the SARS-CoV-2 transmission compared to other neighboring countries [29]. The authors conclude that a mask-wearing strategy irrespective of COVID-19 symptoms mitigates infection of susceptible individuals [29]. Another study from Cowling et al. analyzed data from 60 general outpatient clinics and three cross-sectional surveys in Hong Kong [30]. They findings indicate that rapidly implemented public health measures and the compliance of the population mitigated the SARS-CoV-2 outbreak in Hong Kong [30]. The population readily embraced the recommended prevention methods and social distancing, together with restrictions on the borders, while cases and their surrounding contacts were successfully tracked, isolated, and quarantined [30]. According to the authors, these measures controlled the SARS-CoV-2 outbreak more efficiently and reduced socio-economic tensions without strict nationwide closure and lockdown during the COVID-19 outbreak compared to other countries [30]. The measures also reduced the transmission of influenza in the month of February 2020 [30].

High levels of fatality were recorded in advanced age patients with immune systems compromised by diabetes, hypertension, and cardiovascular disease during SARS-CoV, MERS-CoV, and COVID-19 [31]. Deng et al. compared the characteristics of the fatality group of patients with those of the recovered group [31]. Advanced age, dyspnea, comorbidities, low oxygen saturation, high white blood cell (WBC) count, low lymphocytes, and high levels of C-reactive protein (CRP) are significant indicators of mortality [31]. Giang et al. directly related people aged 65 and above with death rates [32]. They found that the measures of social distancing, lockdown, quarantine, and a high-quality health care system with appropriate numbers of health care workers and emergency rooms helped to decrease fatality rates [33]. The quality and preparedness of the health care system in a country defines the thin line between survival and death.

In most COVID-19 confirmed cases, symptoms such as fever, dry cough, fatigue, runny nose, or other upper respiratory symptoms appeared in SARS-CoV and MERS-CoV [34,35,36,37,38]. Distinguishing COVID-19 from other known respiratory tract viruses clinically is a difficult task due to its non-specificity. A precise diagnosis can be carried out by reverse real-time quantitative polymerase chain reaction (RT-qPCR) testing and radiological study of the cases [35,39,40,41,42]. Fast and reliable methods of COVID-19 testing can be life-saving tools to eradicate the virus in Far Eastern countries. To validate this thesis, we compared the fatality numbers in South Korea and Japan during the SARS-CoV-2 pandemic (Figure 2).

The first confirmed COVID-19 case in Japan was announced on 15 January 2020, while South Korea had a confirmed case five days later [18]. The first fatality in Japan occurred on 13 February 2020, while South Korea’s first COVID-19 fatality occurred on 21 February [18]. The total number of fatalities in Japan on 2 June 2020 was 894 within a population of more than 126 million, and South Korea’s was 24 within more than 5 million [18]. The low fatality rate in both countries indicates excellent pandemic control. Nonetheless, the factors relevant to this success should be discussed. The median age in Japan of 48.6 years is the highest globally, and could have led to the highest global fatality rates because of the higher susceptibility of elderly [28]. The number of deaths is low compared to Mexico with a similarly sized population (>126 million) and 10,167 fatalities until 2 June 2020 [18]. The median age in South Korea is 43.2 [28]. Both South Korean and Japan adopted early intervention strategies to control the viral spread with measures of social distancing and widespread COVID-19 testing, combined with strict follow-up of clusters of infections [43]. Both societies gained experience during the SARS outbreak, whereby they developed the habit of wearing masks, which prepared them for this pandemic. South Korea introduced a dynamic and quick response to the needs of the time. The measurements were implemented in harmony with the public through open information sharing and transparency, thus ensuring public support and collaboration [43]. These joint efforts orchestrated a supply of masks, their distribution, and smart use [43]. A drive-through screening system was used to support general widespread COVID-19 testing and all individuals were followed up via personal tracking [44]. As a result, the pandemic was controlled without a complete lockdown in both countries mainly by widespread COVID-19 testing, individual tracking, and follow-up of clusters. Public support was essential through social distancing, wearing of masks, personal hygiene, transparency during the pandemic, collaboration to meet the needs of the market in mask production, and, most importantly, widespread COVID-19 testing with strict follow up mechanisms for clusters.

Many countries responded with different control mechanisms compared to the experiences of China, South Korea, Japan, and Hong Kong [29,30,44,45]. Highly industrialized, European countries were expected to best cope with the pandemic due to their level of sustainable development. To verify this thesis, we analyzed the data of selected European countries and grouped them according to their population size and/or number of fatalities [18]. Germany (population 2018: 82,927,922) and Turkey (population 2018: 82,319,724) faced their first reported COVID-19 case on 28 January 2020 and 12 March 2020, respectively [18]. Until 2 June 2020, the number of confirmed cases in Germany was 182,028, while Turkey had 164,769 cases. Germany had reported a total number of 8522 fatalities, while Turkey recorded 4563 fatal cases (Figure 3) [18].

The COVID-19 fatality numbers in Germany are almost double the number of fatalities in Turkey. The reason may be due to the higher median age in Germany [24]. The median age of the German population is the second highest after Japan [24]. The arrival of the pandemic in Turkey was more than one month later than in Germany. Both countries reacted after the first COVID-19 fatality with a partial lockdown and social distancing measures, and closure of workplaces, schools, and universities [19]. The borders were closed for non-essential travel [19]. Citizens stranded in other countries were repatriated and quarantined for two weeks. All border crossings were controlled and checked. Authorities started with widespread testing and followed-up on clusters and asymptomatic cases. Turkey started producing masks, effective personal protection equipment (PPE), and respirators to cope with demand inside and outside the country. According to the curves in Figure 4, the peak of the pandemic is over for both countries. Both countries were well prepared to handle the pandemic due to well-developed health care systems and by taking the needed measures to control the spread of the virus. Italy and Spain have younger populations compared with Germany [24]. The median age in Germany is 47.8, while in Italy it is 46.6 and in Spain it is 43.9 [24]. Furthermore, the mean serum vitamin D levels for the populations in Italy and Spain are similar to those of Germany [27]. These factors should have resulted in lower SARS-CoV-2 susceptibility and better control of the outbreak. However, Italy and Spain were confronted with a massive outbreak of COVID-19 in 2020. Italy reported the first confirmed case in 31 January 2020 and the first fatality on 23 February 2020 [18]. Spain confirmed the first case on 1 February 2020 and the first fatality occurred almost one month later on 5 March 2020 [18]. Spain had recorded 239,638 cases and 27,940 fatal cases with a population of approximately 46,723,749 [18]. The number of COVID-19 cases in Italy was 233,197 with a population size of 60,431,283 and 33,475 fatalities until 2 June of 2020 (Figure 4) [18].

The highest fatalities were recorded towards the end of March in Italy, while the number of confirmed cases started to decline 10 days earlier (Figure 4). Widespread COVID-19 testing was initiated in the beginning of March with a delay of one month after the first confirmed case. Both curves descended slowly towards 2 June 2020 which indicates that the SARS-CoV-2 outbreak was under control prior to the summer months (Figure 4). The delay of widespread COVID-19 testing for one month aggravated the pandemic in Italy. Due to this delay, the implemented state of emergency, nationwide closures, lockdown, and further measures did not effectively control the spread of the virus. In addition, lower median age and similar serum vitamin D levels compared to Germany did not have any positive impact on the COVID-19 disease course and did not reduce fatality numbers in Italy.

France and the United Kingdom (UK) were also severely hit by the COVID-19 pandemic. Both countries have around 67 million inhabitants [18]. The total number of fatalities in France was 28,833 and in the UK 39,045 until 2 June 2020 (Figure 5) [18].

France recorded 152,091 and UK reported 276,332 COVID-19 cases until the same date [18]. The highest number of fatalities were counted in France between 31 March and 20 April 2020. UK registered a steady decline of COVID-19 related fatalities after 20 April 2020. According to Ilie et al., the mean serum vitamin D level of the French population is 60 nmol/L, which is higher than that for the populations of Portugal, Spain, Switzerland, UK, Italy, Germany, and Turkey, with 39, 42.5, 46, 47.4, 50, 50.1 and 51.8, respectively [27]. As a result, low serum vitamin D levels have no direct impact on COVID-19 fatality rates. The COVID-19 case and mortality counts on the European continent reveal mixed results that do not depend on temperature, mean age, or mean serum 25-hydroxyvitamin D (25(OH)D) levels, but on the preparedness of the health care system and suitable public health strategies.

The next epicenter evolved on the American continent. The first confirmed case of SARS-CoV-2 infection in the United States of America (USA) was reported on 21 January 2020 one day after that of South Korea [18]. The fatality rate on 10 April 2020 was 2% for South Korea and 3.6% in the USA [46]. South Korea, due to its previous SARS experience, reacted swiftly with sophisticated, new measures including transparent, community-supported policies and widespread COVID-19 testing with strict individual follow-up tracking mechanisms of clusters of transmission [43,44]. The USA had recorded 105,147 total COVID-19-related fatalities and 1,811,277 confirmed cases as at 2 June 2020 (Figure 6) [18]. Figure 6a shows a sharp increase in the confirmed COVID-19 cases after 17 March 2020.

The fatality curve in Figure 6b shows a peak on 15 April 2020. Thereafter, a steady decline in fatality numbers was reported. This curve may indicate that the worst of the outbreak in the USA is over and the peak lasted for around one month after the first reported case of COVID-19 on 17 March 2020. The US started to test for COVID-19 one month after this first confirmed case [46]. This delay of testing caused a rapid viral spread through the population because the infected cases were not identified and followed-up [46]. The unexpected mass of severe cases strained the unprepared health care system. Other compounding factors, such as a lack of drugs, treatment protocols, or vaccines resulted in a health disaster for the USA (105,147) (fatalities in USA due to COVID-19 at the time of writing), in addition to countries such as the UK (39,045), Italy (33,475), Brazil (29,937), France (28,833), Spain (27,940), and Mexico (10,167), which had recorded mortalities in excess of 10,000 prior to 2 June 2020 [18]. Asymptomatic cases are a leading cause of the viral outbreak [46]. Widespread and random testing are effective methods to identify clusters of infections and mitigate the transmission patterns through follow-up and quarantine [46]. Re-testing cases during the isolation and post-quarantine is essential [46].

The first COVID-19 case in Latin America was reported on 26 February 2020 [47]. Subsequently, SARS-CoV-2 spread at a rapid rate across the continent. We analyzed the data for selected countries in Latin America and compared their number of fatalities from the European Centers for Disease Control and Prevention (EDCD) (Figure 7) [18].

Brazil has reported the highest number of fatalities with 29,937, followed by Mexico with 10,167, Peru with 4634, and Ecuador with 3394 [18]. At the present time, it appears that Brazil and Mexico have reached the peak of the pandemic according to Figure 4, while Ecuador and Peru may still be in the initial phase of the pandemic if the curve continues to rise [18]. The COVID-19 pandemic in Ecuador may also overlap with further viral outbreaks of dengue and/or zika, when the country already lacks a suitable number of intensive care unit beds [47]. The arrival of further epidemics such as measles and malaria may exacerbate the burden on the already depleted health care system in Latin America [47]. The situation in South America is far from resolved and could develop into the worst COVID-19 outbreak this year.

China, South Korea, Singapore, Hong Kong, and Taiwan are among the countries that were able to control the COVID-19 outbreak by reducing infection numbers [41,48,49,50,51]. Nationwide closures, lockdowns, travel restrictions, personal hygiene, and social and workplace distancing may help to prevent infection. A modeling study from Singapore investigated the impact of control measures implemented to reduce the spread of COVID-19 [49]. The most effective method was a combination of isolating infected cases, strict quarantine, and closure of schools and workplaces [16,49]. The lockdown of educational establishments and offices (workplace distancing) paved the way for online solutions, which helped to education and work to continue from home. After the outbreak in Wuhan, China, the Korean government activated a 24/7 emergency response system to screen all travelers entering the country from that city [49]. In Taiwan, proactive and comprehensive health checks of passengers from Hubei province were established quickly from the beginning of January 2020 [50]. The Taiwanese Centers of Disease Control (CDC) tested 2105 cases by February 28 using multiplex PCR analysis with FilmArray^TM^ Respiratory Panel and confirmed 34 COVID-19 patients [50,51,52]. According to Hsih et al., COVID-19 is more contagious than seasonal respiratory pathogens but infected cases have common clinical and laboratory results [52]. Therefore, each suspected case needs close follow-up through individual tracking, gathering of mobility and contact data, isolation, and quarantine [52]. Hsih et al. underline the elevated pandemic level of SARS-CoV-2 manifested by prolonged viral shedding, and the large number of asymptomatic and mild illness cases that remain undiscovered and continue to spread COVID-19 [52]. They also note that some recovered patients still had detectable virus levels for almost two further weeks [52]. Furthermore, some patients showed negative COVID-19 test results from their naso-oropharyngial system but positive results in their sputum or fecal specimens [52]. The authors are concerned about possible SARS-CoV-2 transmission through the fecal–oral route [52].

### AE and Public Health Strategies During COVID-19

We investigated the SARS-CoV-2 pandemic in the United Arab Emirates (UAE) as an example of a highly populated, globally interconnected country with an equatorial hot climate and excellent control of the COVID-19 outbreak. In this work, we present the level of public health responses and control of SARS-CoV-2 in the UAE, and underline that publications about COVID-19 in the UAE are very rare. Figure 8a shows the confirmed cases and Figure 8b the fatalities in the UAE.

The population in the UAE totals around 7 million according to the 2018 census [18]. The total number of confirmed cases was 35,192 and the number of fatalities was 266 on 2 June 2020 [18]. The first fatality occurred on 22 March, almost one month after the first reported case [18]. During the month of March, the median temperature was 23.6 °C and the relative humidity was 59%. This high temperature and low relative humidity were not conducive to SARS-CoV-2 proliferation [24]. Similar climatic conditions also exist in Hong Kong, Singapore, and other Gulf countries [24]. All of these countries are characterized by warm temperatures and low humidity in winter months, dense populations in their capital cities, well-connected international airports, and low median age (with the exception of Hong Kong in the case of the latter) [24]. These countries can be counted as examples of excellent pandemic control.

The UAE authorities responded in a rapid, flexible, effective, and transparent way to curb the spread of SARS-CoV-2. Rapidly implemented public health measures, closure of borders, nationwide closures of educational institutions, complete lockdowns for certain periods, and work-from-home schemes reduced mobility and helped to efficiently control the viral outbreak in the UAE [53]. Via stepwise measures, on 25 March 2020 the UAE General Civil Aviation Authority suspended all flights into and from the country [53]. The “National Disinfection Programme” was implemented by the Ministry of Health and Prevention (MoHaP) and Ministry of Interior (MOI) to disinfect public facilities, public transport, and roads [53]. The disinfection campaign was initiated on 26 March 2020 and performed every night from 8 pm to 6 am [53]. Nationwide COVID-19 testing was launched by the Ministry of Health and Prevention, the Ministry of Interior, and the National Emergency Crisis and Disasters Management Authority at the end of March 2020 [18]. Drive-through COVID-19 testing facilities were opened in many locations throughout the UAE and continue to serve the population together with mobile laboratory units, home testing for people of disabled hospitals, and licensed medical centers [53].

Recommendations for community-wide social distancing and a mask-wearing strategy complemented by gloves and personal hygiene measures were embraced by the population. The beginning of the fasting month of Ramadan eased the closure of restaurants, shops, and other facilities for entertainment and leisure activities. Prayers were suspended shortly before the month of Ramadan, and this suspension was extended on 16 March 2020 by the National Emergency Crisis and Disaster Management Authority and the General Authority of Islamic Affairs and Endowments until further notice. The Ministry of Education announced school and university closures on 8 March 2020 coinciding with the spring break and implemented a distance learning initiative on 22 March 2020 [53]. Education continued without disruption through online teaching after the spring break and is expected to remain distance-learning-based until the end of the educational year in summer 2020 [53]. Remote working was adopted by most institutions of the federal and local governments in the UAE to protect employees and customers. The pre-existing advanced technological infrastructure of the UAE enabled remote working and online meetings throughout the educational and public sectors [53]. Numerous apps, such as ALHOSN UAE, were created to track cases and advise the population [53]. Offices and malls briefly closed and reopened towards the end of May 2020 with strict control measures to prevent COVID-19 spread. The closure of all shopping malls, with the exception of pharmacies, food retail outlets, cooperative societies, grocery stores, and supermarkets, was announced on 25 March 2020 [53]. Children below 12 years and individuals above 60 years were not permitted to enter malls and outlets [53]. Everyone entering any facility is required to wear surgical masks and gloves, and is screened for symptoms of fever. Disinfectant dispensers are widely distributed in all buildings. Most institutions, such as Ajman University in Ajman, UAE, contributed unreservedly to the campaign by disseminating the recommendations and applying the necessary measures in a timely manner (Figure 9).

The COVID-19 outbreak is currently under control in the UAE: the number fatalities is low and shows a downward trend (Figure 8). The health care system was rapidly supported and prepared for a mass outbreak. Steadily rising temperatures from April 2020 and high relative humidity did not prevent COVID-19 from spreading. The curve in Figure 8b peaks on 12 May 2020 in terms of the number of fatalities, while in Figure 8a the peak for confirmed cases appears around 10 days later [18]. Increasing UV irradiation, temperature, and relative humidity may have been supportive factors in controlling the SARS-CoV-2 outbreak in the UAE.

The influence of hot weather and increasing humidity in the coming months should be questioned. The virus cannot be inactivated solely by high temperatures, and can survive in the human body [24]. The outdoor climate is highly interconnected with the indoor climate and human dwelling patterns. With the steady rise of temperatures above 45 °C and high humidity, indoor environments should be included in thinking about the virus. When the pandemic began at the end of February, the temperature was suitable for outdoor activities. In very hot climates, people gather in indoor environments. The summer months are marked by indoor dwelling, and spending time in air-conditioned closed apartments, offices, buildings, and shopping malls. The indoor environment in summer may cause aggravated SARS-CoV-2 transmission patterns and an increase in COVID-19 cases and fatalities. New waves of infection can only be prevented by continuing the community-wide, strict measures of social distancing, personal hygiene, wearing of surgical masks and gloves, nationwide testing, and individual tracking of clusters and their contacts. Equatorial countries may witness an increase in cases and/or fatalities during the coming months. The spread of SARS-CoV-2 in the summer months can be prevented by adding new strategies to the existing measures. Our recommendation for countries with hot climates in summer is to prevent an influx of cases from other countries and to control indoor environments. The latter can be achieved via sufficient ventilation several times a day, continued strict disinfection protocols using antimicrobial agents/disinfectants for surfaces, and use of humidifiers equipped with essential oils and/or antimicrobial plant extracts. Split air conditioning systems may be better than central systems. These air conditioning systems need to be regularly cleaned and maintained. Sterilization of indoor environments may be possible using UV lamps as an alternative. These measures may inactivate the virus and reduce the number of viral droplets in indoor air. The temperature in closed rooms may not reach 4–12 degrees Celsius, but virus-loaded droplets can easily infect anyone in the vicinity by entering the naso-pharyngial tract or the eyes if the viral density in the air is high, or through contact transmission by fomites [16,17,24]. However, airborne transmission is a controversial issue, which depends on climatic pre-existing conditions (indoor/outdoor environments) and needs a case-by-case approach when public health measures are to be implemented. According to Cheng et al., a community-wide mask-wearing strategy may decrease the shedding of respiratory droplets or saliva from symptomatic or asymptomatic COVID-19 cases [29]. Pecho-Silva et al. indicate that airborne transmission can occur due to floating viral droplets suspended in the air [54]. They suggest that droplet nuclei may be only produced by specific medical treatment protocols such as intubation and nebulization [54]. Stadnytskyi et al. reported that SARS-CoV-2-loaded droplet nuclei can be transmitted from an asymptomatic case while speaking in a closed room with stagnant air [55].

As a result, transmission by fomites and direct inhalation of droplets in indoor environments is a serious problem in closed, unventilated, and crowded rooms with very poor air exchange. Although the issue remains controversial, strict guidelines should be followed to prevent viral infections until further research emerges. India implemented strict nationwide lockdowns to address the problem of viral transmission; however, by keeping the population indoors it may have created precisely these unventilated, crowded room scenarios that can lead to a high number of cases and fatalities.

The Indian government reacted rapidly with very strict nationwide closures and a complete lockdown, and prevented an uncontrolled SARS-CoV-2 outbreak from March to June 2020 [56]. The first fatality in India was recorded on 13 March 2020 (Figure 10) [18].

The first COVID-19 case appeared with a delay of more than one month on 30 January 2020 [18]. The total number of confirmed cases for India was 198,706 within a population of around 1.4 billion [18]. A total of 5598 fatal cases were reported until 2 June 2020 [18]. The COVID-19 outbreak was controlled firmly until beginning of April 2020, however, incidence thereafter began to increase. The curve currently shows an upward trend in fatalities and a peak is not yet visible in Figure 10. A lack of awareness and a relaxed attitude has resulted in an increase in COVID-19 cases and fatalities [56]. The situation in India is far from over and strict regulations should continue, paired with mask-wearing strategies, nationwide testing, and control of overcrowded indoor environments.

We collected and analyzed the implemented public health measures during the SARS-CoV-2 pandemic. Different countries implemented various measures and experienced differing results during the COVID-19 pandemic (Table 1).

## 4. Conclusion and Recommendations

A lack of preparedness for the sudden onset of COVID-19 by governments and health care systems, combined with inadequate public health strategies and deficiencies in diagnostic mechanisms, treatment options, and management protocols, aggravated the virulence of SARS-CoV-2 globally. There has been a direct impact of public health strategies on case count and mortality. Far Eastern countries with experience of the SARS outbreak controlled the pandemic with the lowest number of COVID-19 fatalities. COVID-19 case and mortality counts in Europe are not directly dependent on temperature, mean age, or mean serum 25-hydroxyvitamin D (25(OH)D) levels, but on preparedness of countries’ health care systems and adequate public health strategies.

Infection control measures in indoor environments are essential in viral transmission prevention and need to be taken into consideration. Indoor environments are interconnected with climatic conditions. Indoor climates are crucial to the control of the SARS-CoV-2 outbreak. Development of health care systems by increasing the number of hospitals, critical care units, and health care personnel are also key factors, and are particularly needed for the protection of the vulnerable and elderly population. The preparedness of the health care system includes stockpiles of appropriate and effective personal protection equipment (PPE), fast and reliable testing methods, and cluster and individual tracking of cases and their contacts. Recommendations of social distancing, personal hygiene, avoiding of gatherings, and mask wearing must be strictly followed even in indoor environments and household settings, particularly for the protection of those aged over 65 years. Further COVID-19 outbreaks may be expected due to new waves and mutations of the virus. Rapidly implemented, transparent region-specific public health strategies and community-wide compliance are essential during the COVID pandemic and the post-lockdown period. Increased health literacy in the population can improve the management and control of further global pandemics.

## Figures and Tables

**Figure 1 ijerph-17-05616-f001:**
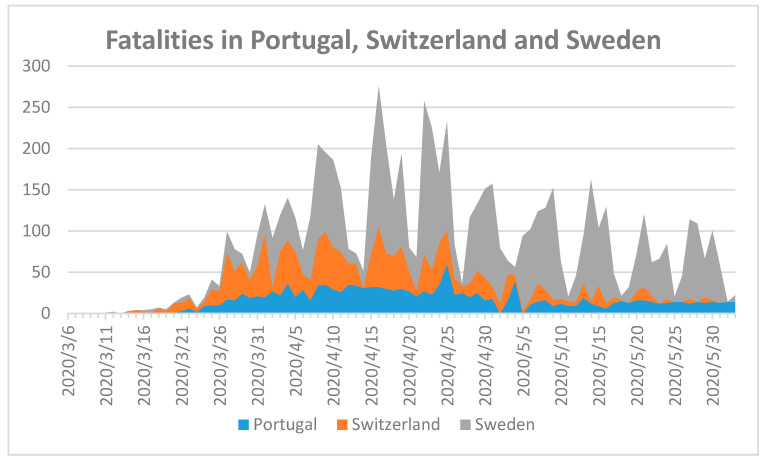
Fatalities during COVID-19 in Portugal, Switzerland, and Sweden until 2 June 2020 [18]. First fatality occurred in Portugal on 18 March, Switzerland on 6 March, and Sweden on 12 March 2020.

**Figure 2 ijerph-17-05616-f002:**
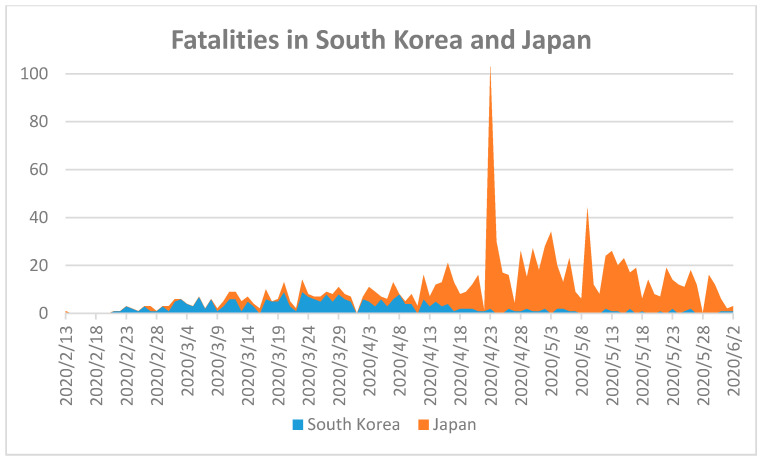
Fatalities during COVID-19 in South Korea and Japan until 2 June 2020 [18]. First fatality in South Korea on 21 February 2020 and Japan on 13 February 2020.

**Figure 3 ijerph-17-05616-f003:**
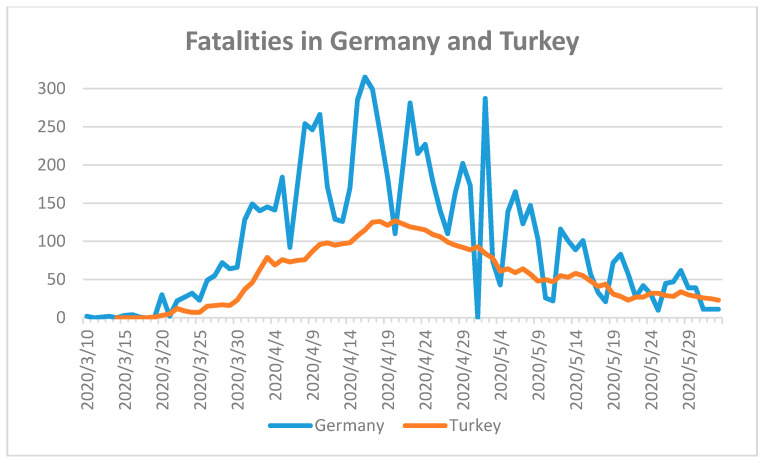
Fatalities during COVID-19 in Germany and Turkey until 2 June 2020 [18]. First fatality in Germany on 10 March 2020 and Turkey on 19 March 2020.

**Figure 4 ijerph-17-05616-f004:**
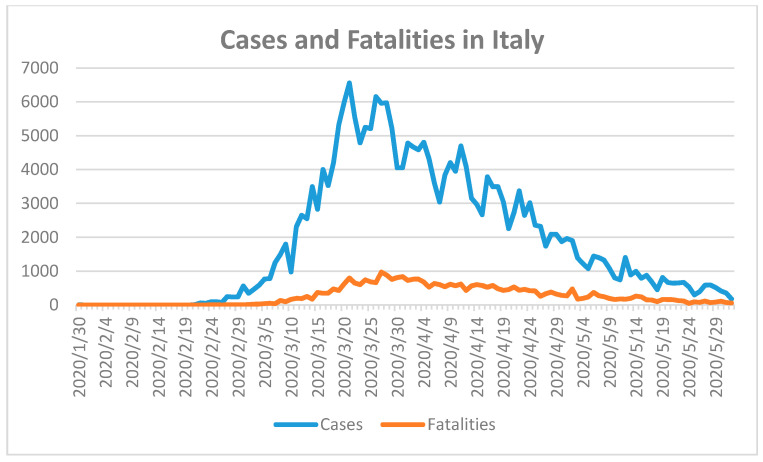
Fatalities during COVID-19 in Italy until 2 June 2020 [18]. First confirmed case in Italy on 31 January 2020 and first two fatal cases on 23 February 2020.

**Figure 5 ijerph-17-05616-f005:**
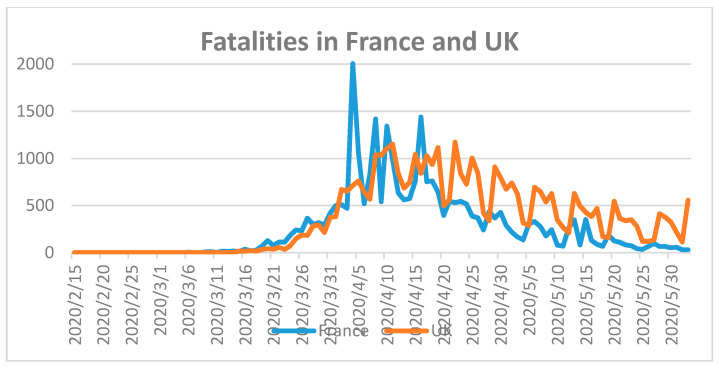
Fatalities during COVID-19 in France and UK until 2 June 2020 [18]. First fatality in France on 15 February 2020 and in UK on 7 March 2020.

**Figure 6 ijerph-17-05616-f006:**
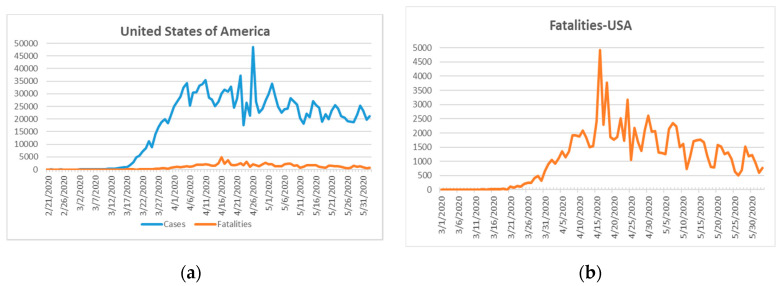
Confirmed COVID-19 cases/fatalities in the USA until 2 June 2020 [18]. First fatality in US on 1 March 2020. From left to right: (**a**) confirmed cases and fatalities in the USA; (**b**) fatalities in the USA.

**Figure 7 ijerph-17-05616-f007:**
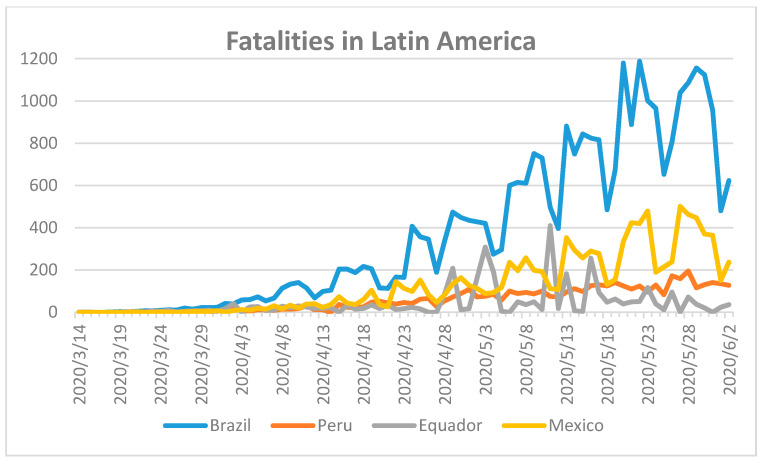
Fatalities during COVID-19 in Latin America until 2 June 2020 [18]. First fatality in Brazil on 18 March, Peru on 20 March, Ecuador on 14 March, and Mexico on 24 March 2020.

**Figure 8 ijerph-17-05616-f008:**
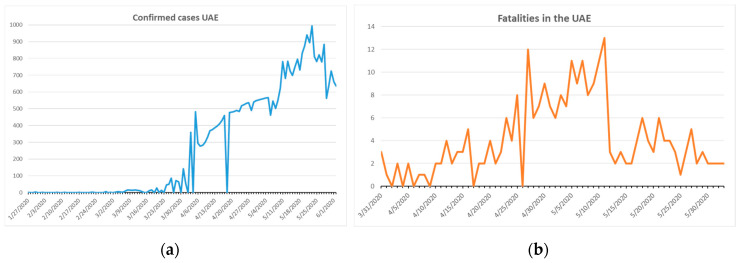
Confirmed COVID-19 cases and fatalities in the UAE until 2 June 2020 [18]. First confirmed case on 27 January 2020, first fatality on 22 March 2020 in UAE. From left to right: (**a**) confirmed cases in the UAE; (**b**) fatalities in the UAE.

**Figure 9 ijerph-17-05616-f009:**
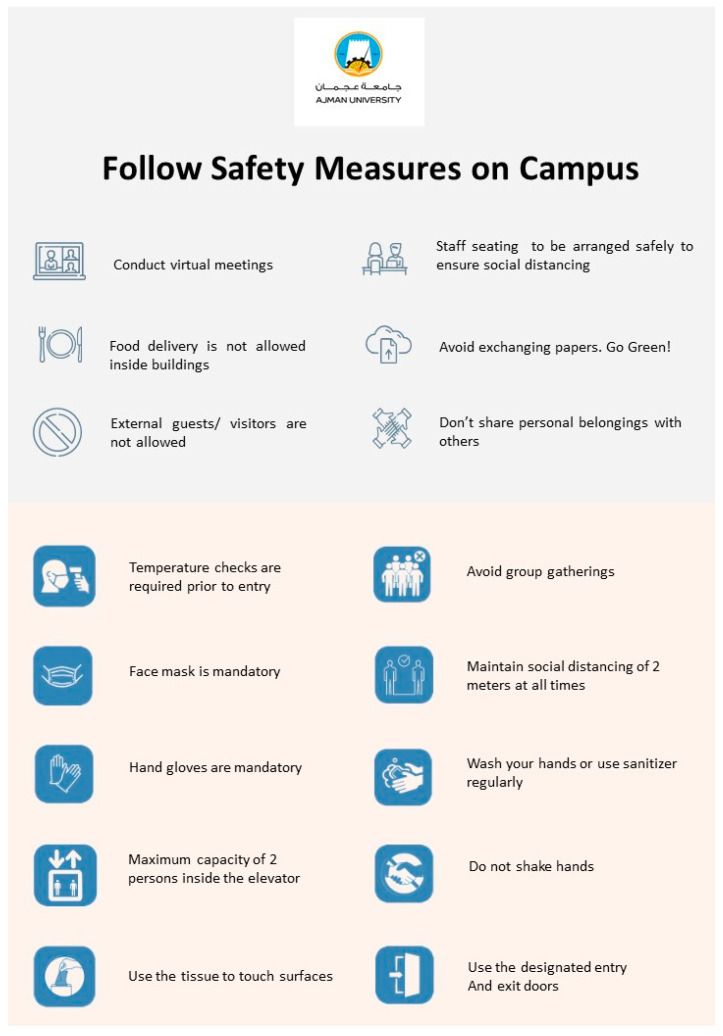
Safety recommendations and measurements during the COVID-19 pandemic implemented by Ajman University (AU), Ajman, UAE. (Reproduced with agreement of the AU Department of Environmental Health and Safety (EHS) on 11 May 2020.).

**Figure 10 ijerph-17-05616-f010:**
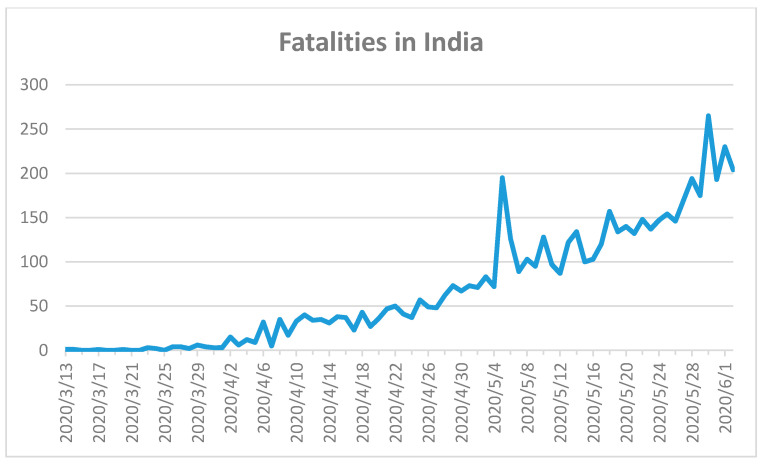
Confirmed COVID-19 fatalities in India until 2 June 2020 [18]. First fatality on 13 March 2020 in India.

**Table 1 ijerph-17-05616-t001:** Countries, their markers, some public health measures, and outcomes during the COVID-19 pandemic.

No.	Countries [24]	Pre-Existing Conditions [19]	Preventive Measures	Results [18,24]
1	Taiwan [52], South Korea [43,44,51], Hong Kong [29,30], Singapore [29,49], Japan [43]	High population density. Old age population in Japan. Prepared health care system. Culture of social-distancing. Previous experience with SARS.	Rapid implementation of public health measurements with community-based support. Closures, social distancing, personal hygiene, masks-on strategy. Nationwide and random COVID-19 testing with different types of methods. Personal tracking of cases and clusters, follow-up, isolation, quarantine, and re-testing during quarantine.	Under Control (Figure 2)
2	UAE [53]	High population density in urban areas. Developed health care system. Rising temperatures and relative humidity.	Nationwide closure. Lockdown on cities. Rapid implementation of public health measurements with community-based support. Closures, social distancing, personal hygiene, masks-on strategy. Nationwide and random COVID-19 testing by different types of methods. Personal tracking of cases and clusters, follow-up, isolation, quarantine, and re-testing during quarantine. Educational institutes closed quickly. Online teaching implemented in schools and universities, like in Ajman University, Ajman, UAE. Work from home. Public transportation stopped. Sanitizing huge areas in cities during nights. Availability of personal protection equipment and other items of medical significance ensured by government. Establishment of services to prevent spreading COVID-19 (such as Dawak Li Darek in Ajman, UAE, delivering chronic patients the needed drugs to their doorsteps.) Initial closure of malls except supermarkets and pharmacies. Guidelines for outlets permitted to receive customers after opening [53]: limited entry of customers to 30 percent of its capacity maintained distance of at least two meters between customers no crowding allowed Recommendations to the public [53]: observe physical/social distancing wear face masks and gloves outdoors stay at home (unless needed for purchase of food, medicine or to receive medical care or being employed in vital sectors Follow the guidelines on family visits follow medical advice issued by relevant authorities perform prayers at home do not enter shopping malls and outlets if above 60 years or below 12 years	Under Control (Figure 8 and Figure 9)
3	Germany, Turkey	Old age population in Germany. High population density in urban areas. Prepared health care system.	Quick closure of educational institutes. Lockdown on time. Work from home. Rapid implementation of public health measurements with community-based support. Closures, social distancing, personal hygiene, masks-on strategy. Nationwide and random COVID-19 testing with different types of methods. Personal tracking of cases and clusters, follow-up, isolation, quarantine, and re-testing during quarantine.	Under control (Figure 3)
4	Italy [51], Spain	Old age population. Low serum 25-hydroxyvitamin D (25(OH)D) concentrations in elderly population.	Late lockdown. Lack of harsh preventive measures. Delayed implementation of mobility restrictions. Closures, social distancing, personal hygiene, masks-on strategy. Late nationwide and random COVID-19 testing. Late tracking of cases and clusters, follow-up, isolation, quarantine, and re-testing during quarantine. Unprepared, overwhelmed health care system due to major outbreak.	Hit hard, under control now (Figure 4)
5	USA [46], UK	Unprepared, overwhelmed health care system due to major outbreak.	Delayed lockdown. Lack of harsh preventive measures. Lack of prevention strategy. Late ban on mobility. Late nationwide and random COVID-19 testing. Late personal tracking of cases and clusters, follow-up, isolation, quarantine, and re-testing during quarantine.	Hit hard, ongoing outbreak (Figure 5 and Figure 6)
6	Sweden [27]	Prepared health care system.	No lockdown	Ongoing outbreak (Figure 1)
7	Latin America [47,51]	Unprepared or poor health care system. Overlap with epidemics (Dengue, Malaria, Measles, Zika)	Delayed lockdown. Lack of harsh preventive measures. Lack of prevention strategy. Late ban on mobility.	Hit hard (Figure 7)
8	India [56]	Unprepared health care system. High population density in urban areas.	Work from home. All services except emergency services locked down. Janata curfew. Lockdown in metropolitan cities.	Hardly under control (Figure 9)

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
