# Peer review of "A Look Behind the Scenes at COVID-19: National Strategies of Infection Control and Their Impact on Mortality"

_ijerph, 2020, doi:10.3390/ijerph17155616_

Round 1

Reviewer 1 Report

Dear authors, most of suggestions have been successfully addressed. In particular the manuscript has beensignificantly shortened, alla redundant parts have been eliminated and, overall, its clariy is improved. Now the final aim of the paper is clear.

Despite it is still a bit too long, I think that the paper could be accepted in the current format.

Reviewer 2 Report

Dear authors, thank you for your effort to imporve the work. In the current format it is very more clear, better structuredm and interesting. I suggest to accept the paper in the current format.

This manuscript is a resubmission of an earlier submission. The following is a list of the peer review reports and author responses from that submission.

Round 1

Reviewer 1 Report

Dear authors, I read your paper. Some interesting ideas are presented, and some issues are described in details. However, many important flaws are present: it is overall confused, too much long, full of repetitions, and sometimes not scientifically well-sounding. In this form, it is not suitable for publication on IJERPH.
Some general considerations:
- It is not clear which kind of paper it is. Not a systematic review, as the search strategy is “open” and poorly defined, and the articles cited are selected on the basis of authors' opinions and selection criteria, and not on a pre-defined search criteria (as in a systematic review should be). It is not a research article, as no original data is presented, and the authors’ findings on some topics of interest (use of nanomaterial, plant-based essential oils, the so-called “Floyd effect”) are mainly authors’ opinions and suggestions, supported by scarce or not applicable scientific data (for example, data about nanomaterials mainly refer to bacterial infections, and no data are available about COVID-19);
- What is the main topic or topics? An overview of COVID-19 is initially presented, covering various topics. In particular, the impact of public health measures on the number of cases and mortality is discussed, comparing data from various countries. In addition, various other sub-topics are covered: clinical presentation, diagnosis, treatment. This part of the article is largely redundant (the same concepts are repeated in many different parts), while on other sub-topics the data provided is largely incomplete (especially in treatments, where tolicizumab and other drugs, such as Low-molecular-weight heparin are not cited). Then, some more "original" topics emerge in various parts of the article, in particular the role of vitamin D, the role of the household transmission, the so-called "Floyd Effect", the antibiotic resistance and nanomaterials, the use of essential oils. These arguments are very different from one another, and are difficult to understand in the current structure of the article.

Specific comments:

  • The title is confusing, and refers to different unrelated parts of the paper;
  • The abstract reflects the flaws in the article: it is long, not well-structured, the aim of the paper is unclear, some affirmations are based on literature data and other reflects authors’ opinions and personal ideas not supported by data;
  • The sentence on lines 43-45 is wrong: the virus was not called 2019-nCoV, then SARS-CoV-2 and finally COVID-19. COVID-19 is the name of the disease, while the virus name changed from 2019-nCoV to SARS-CoV-2;
  • Many data presented in introduction, for example virologic and taxonomic data, are not useful for article purposes, and should be deleted. In general, the article should be reduced by at least half, eliminating all the repeated and not useful parts for the main topics;
  • Considering that one of the main topic of the paper is the containment and public health strategy, instead of details about virology, data about modes of transmission should be included. Is this virus airborne-transmitted? Which are the most relevant modes of transmission (airborne, droplets, fomites, aerosol, direct contacts)? About this topic a large literature exists, and in your paper it is only marginally descripted;
  • Many parts of your paper is dedicated to Vitamin D deficiency, and its correlation with negative COVID-19 outcome. This association is suggested by few data, and in my opinion it is not possible to base too many reflections and inferences on such scarce data;
  • In the results, you usually present general data about one or more countries, trying to answer to a specific questions. Often, data presented do not answer to the specific question. For example, after reading, I’m still asking: Which is the role of Vit.D? Are strict closures and complete lockdowns necessary? Have industrialized countries better opportunities to respond to COVID-19? Which is the role of dry and hot climate in COVID-19 control?
  • Moreover, you usually present sounding data. But, at the end of many paragraph, a personal opinion, not based on any data, is included. For example, sentences as: “Ventilation can be also supported by split air-conditioning units in rooms, humidifiers and diffusers with essential oils and UV lamps.” or “Masks and PPE with new antimicrobial, self-disinfecting coating have the potential to diminish viral transmission.” should not be included in the Results, but in Discussion only, since they are opinions, and not facts;
  • Table 1 is just a repetition of text, with no additional data. Then, according to the structure, it seems that any presenting symptom is coupled with a complication (I supposed it on the basis that ARDS is cited three times): if yes, why kidney failure is the complication of fever? Why Acute Hearth Failure is the complication of fatigue? I suggest to remove the table at all;
  • Sections about symptoms, diagnosis and treatment are in general incomplete, and moreover, they are completely off-topic in the paper. I suggest removing them completely;
  • Table 2 is difficult to understand, also. The choice to put together in the same column the pre-existing conditions (prepared/unprepared health system; median age) with the response strategy is confusing. I suggest to add a column, and list separately the pre-existing conditions and the response strategy;
  • The part about psychological impact and the so-called “Floyd effect” is very interesting but, as already said for other things, it is off-topic respect to the main arguments of the paper. I think that your opinion is interesting and respectable, but this paper is not the right place to insert this topic;
  • The section about Antimicrobial resistance and nanomaterial is poorly scientifically sounding. Most of references refer to bacterial diseases, and many papers are about the reinforcement of immune system. No evidence exists that these approaches may have a role in COVID-19 outbreak, and in some case even the biological plausibility is lacking or very weak. In particular about the reinforcement of immune system: are we sure that this approach is valid in COVID-19? Given that most of severe symptoms are driven by a hyper-inflammatory response, and that an emerging positive role is being given to corticosteroids as modulators of immune response, are we sure that improving immune response is good?

Overall, I suggest to completely revise the paper, avoid redundant contents, and to split it in different papers. The main one may include the discussion about different national strategies and their impact on case count and mortality. Other papers may be on psychological effects and on role of nanoparticles. These arguments are very different among them and cannot be presented in the same article.

Reviewer 2 Report

Dear authors, many interesting reflections and opinions are presented in your paper. Especially the analysis of national data, compared with national response strategies, is remarkable. However, these interesting data are included in a too-long, not well-organized paper. Many different arguments are presented, quite different among them. The logical connection between different parts is lacking.

The general impression is that the authors have opinions, in some cases even shareable and interesting, such as the need for infection control measures in indoor environments, on very different and unrelated topics. These topics need, for each of them, a specific handling, more supported by data, although not necessarily original data produced by the author. Instead, all of these topics flowed into an overall article, in which they are confusing.

Specific comments:

  • The title does not reflect the paper arguments, please revise it;
  • COVID-19 is the disease name, and not virus name, that changed one time only;
  • Please state more clearly the aim of the paper;
  • You sometimes refer to role of asymptomatic transmission. Please consider that, until now, the role of completely asymptomatic patients in transmission is still unclear and debated. Instead, the role of pre-symptomatic transmission is well defined (I mean, those who develop symptoms, start to disseminate the virus 1-2 days before the symptom onset). We can not refer to pre-symptomatic as asymptomatic, since these are different concepts;
  • Table 1 is difficult to read. Do you mean that complications listed are related to each presenting symptom? If yes, please specify and give a reason for coupling symptoms and complications;
  • Table 3 is too long, and includes many indications not discussed in the text. Please revise it limiting to topics included in the paper.

Reviewer 3 Report

Although the manuscript is well written, it is long-winded and it doesn't give any additional contribution to scientific progress in the field of COVID-19 research. A review should be focused on a specific field to achieve the dignity of pubblication, while the present manuscript appears to be a melting pot of several COVID-related issues, none of them supported by strong evidence. I recommend to extensively cut and revise the manuscript, focusing the dicussion on a single issue (for example nanomaterials), before to reconsider the submission.